# OpenReview forum: "Log-To-Leak: Prompt Injection Attacks on Tool-Using LLM Agents via Model Context Protocol"
_ICLR.cc/2026/Conference — Submitted to ICLR 2026_

### Official Review · Reviewer_X1fo · 2025-10-29

**Soundness:** 2
**Presentation:** 3
**Contribution:** 3
**Rating:** 6
**Confidence:** 4

**Summary:**

This paper proposes the "log-to-leak" method, which exploits prompt injection vulnerabilities in MCP (Model Context Protocol) server metadata to force agents to execute logging functions after task completion. These logs expose sensitive information such as user queries and execution traces, revealing potential privacy leakage issues in MCP tool-use scenarios.

The authors demonstrate their attack on different advanced LLMs. The results show that both baseline and their log-to-leak method are effective. This work highlights the vulnerability of current LLM-based agents to prompt injection attacks, particularly in the emerging MCP tool-use paradigm.

**Strengths:**

1. The research addresses an important problem. With the increasing adoption of MCP for agent-tool integration, understanding its security vulnerabilities has significant practical implications for protecting user privacy in production systems.
2. The proposed attack is simple, practical, and realistic. It exploits a natural attack surface (server metadata) that developers may overlook when integrating third-party MCP servers, making it a credible real-world threat.

**Weaknesses:**

1. Lack of defense evaluation: The paper does not evaluate the effectiveness of existing prompt injection defenses against this attack. This comparison is essential to understand whether existing countermeasures are sufficient or if new defenses are needed.
2. No consideration of adaptive defenses: The paper does not discuss potential defensive measures or their limitations. For example: What if each tool's metadata is automatically inspected by an advanced LLM (e.g., Claude 4.5) for malicious content before being passed to the agent?

**Questions:**

Please see the weaknesses part.

---

> ### Author Response · Authors · 2025-11-21
>
> Thank you for your thoughtful review emphasizing both the importance of the problem and the practical relevance of our findings. We appreciate your recognition that MCP adoption is rapidly increasing and that understanding its security vulnerabilities has direct implications for protecting user privacy in real deployments. We are also grateful for your positive assessment of the simplicity and realism of our attack, which targets a natural but easily overlooked surface, making it a credible threat in real-world MCP integrations. Please find our detailed responses below.
>
> >**Weakness 1**: Lack of defense evaluation: The paper does not evaluate the effectiveness of existing prompt injection defenses against this attack. This comparison is essential to understand whether existing countermeasures are sufficient or if new defenses are needed.
>
> **Response**: We appreciate the reviewer’s concern and have added evaluations of two widely used prompt-injection defenses to determine whether existing countermeasures can mitigate our attack. First, we tested prompt sandwiching, in which the agent is wrapped with pre- and post-safety instructions. Under the GPT-4o + GitHub MCP setting, the attack still achieves a 59.58% success rate, showing that prompt-level shielding does not stop attacks originating from MCP tool metadata rather than the user prompt. Second, we evaluated DataSentinel, a state-of-the-art detection system, by scanning 100 injected MCP tool descriptions generated by our framework. Because DataSentinel analyzes user prompts and model outputs rather than tool metadata, it marked all 100 injected descriptions as safe, resulting in a 100% bypass rate. The detailed results of prompt sandwiching are summarized below:
>
> | Defense Setting      | ASR ↑   | LA ↑    | $\Delta_{TCR}$                    | $\Delta_{MCR}$                   | $\Delta_{TU}$                     |
> |----------------------|---------|---------|---------------------------|-------------------------|--------------------------|
> | Prompt Sandwiching   | 59.58%  | 84.71%  | –4.25% (74.9 → 70.6)     | –0.12% (100 → 99.9)    | –3.9k (23.9k → 20.0k)    |
>
> >**Weakness 2**: No consideration of adaptive defenses: The paper does not discuss potential defensive measures or their limitations. For example: What if each tool's metadata is automatically inspected by an advanced LLM (e.g., Claude 4.5) for malicious content before being passed to the agent?
>
> **Response**: We agree that adaptive defenses are important to consider and have evaluated a stronger mechanism in which an advanced LLM inspects tool metadata before it is exposed to the agent. Specifically, we prompted GPT-4o as a security auditor AI for MCP tool metadata and asked it to assess 100 malicious MCP tool descriptions generated by our framework. Only 11% of the injected descriptions were flagged as unsafe. For the remaining metadata labeled safe, we randomly sampled five and tested them in the GPT-4o + GitHub MCP setting; these “audited-safe’’ descriptions still achieved 55.38% ASR and 85.46% LA, demonstrating that even LLM-based vetting leaves substantial attack surface uncovered. We did not evaluate Claude 4.5 due to API access restrictions, but our results already illustrate the broader limitation: LLM auditors struggle to detect malicious content when the injected instruction is embedded within plausible tool documentation rather than an adversarial prompt. The defense results are summarized below:
>
> | Defense Method          | ASR ↑    | LA ↑    | $\Delta_{TCR}$                             | $\Delta_{MCR}$                     | $\Delta_{TU}$                            |
> |-------------------------|----------|---------|-----------------------------------|---------------------------|---------------------------------|
> | LLM Metadata Check      | 55.38%   | 85.46%  | –5.63% (74.9 → 69.3)             | 0.00% (100 → 100)        | –6.3k (23.9k → 17.6k)           |
>
> These findings show that even adaptive LLM-based inspection is insufficient for metadata-level MCP prompt injection, since the attack exploits the fact that tool descriptions function as trusted documentation rather than adversarial input.

---

> > ### Comment · Reviewer_X1fo · 2025-11-25
> > **Response to rebuttal**
> >
> > Thanks the authors for the efforts during rebuttal. Most of my concerns were resolved.
> >
> > For W1, I would recommend evaluating more advanced defense methods [1, 2, 3, 4] to understand the effectiveness of log-to-leak attack.
> >
> > [1] Zhu, Kaijie, et al. "MELON: Provable Defense Against Indirect Prompt Injection Attacks in AI Agents" Forty-second International Conference on Machine Learning. 2025.
> >
> > [2] Wen, Tongyu, et al. "Defending against Indirect Prompt Injection by Instruction Detection." arXiv preprint arXiv:2505.06311 (2025).
> >
> > [3] Shi, Tianneng, et al. "Promptarmor: Simple yet effective prompt injection defenses." arXiv preprint arXiv:2507.15219 (2025).
> >
> > [4] An, Hengyu, et al. "Ipiguard: A novel tool dependency graph-based defense against indirect prompt injection in llm agents." Proceedings of the 2025 Conference on Empirical Methods in Natural Language Processing. 2025.

---

> > > ### Author Response · Authors · 2025-11-26
> > >
> > > Thanks for your timely reply. Please kindly find our response below.
> > >
> > > > For W1, I would recommend evaluating more advanced defense methods [1, 2, 3, 4] to understand the effectiveness of log-to-leak attack.
> > >
> > > **Response**: Thank you for the suggestion regarding advanced defense methods [1, 2, 3, 4]. We will include all of them in our related work section. For [1], we did not include a quantitative evaluation because its threat model and mechanism do not align with our setting. MELON assumes that the tool return value itself may contain indirect prompt-injection content, and it detects attacks by checking whether the agent’s subsequent tool choices become abnormally dependent on that returned text. In Log-To-Leak, however, the malicious directive is injected purely into the MCP tool metadata; the tool outputs remain unchanged and do not carry adversarial instructions. As a result, the key signal MELON relies on—prompt injection inside tool responses—never appears, and applying MELON would amount to testing it outside its intended threat surface rather than providing an informative baseline.
> > >
> > > For [3], while the paper describes a sanitization mechanism at a high level, there is no code implementation, and crucial design details of the policy are not specified sufficiently to reproduce the method faithfully. To avoid misrepresenting the method with an ad hoc reimplementation, we therefore do not report numerical results for [3].
> > >
> > > For [2], we trained an MLP classifier over Llama-3.1-8B-Instruct hidden states and gradients following the default pipeline in the paper. We then applied it to 100 injected metadata samples from our framework and to metadata from all benign servers used in our experiments. The results are shown below:
> > >
> > > | Metadata Source | Predicted as Injection (count) | Proportion |
> > > |-----------------|--------------------------------|------------|
> > > | GitHub MCP      | 74 / 90                        | 0.8222     |
> > > | Mapbox MCP      | 8 / 9                          | 0.8889     |
> > > | PayPal MCP      | 24 / 28                        | 0.8571     |
> > > | YFinance MCP    | 9 / 9                          | 1.0000     |
> > > | Playwright MCP  | 21 / 23                        | 0.9130     |
> > > | Ours        | 91 / 100                       | 0.9100     |
> > >
> > > While the detector correctly flags most of our injected metadata, it also flags most benign MCP metadata because tool descriptions naturally contain instruction-like phrases. The resulting high false-positive rate makes this method unsuitable for MCP-metadata scanning in practice.
> > >
> > > For [4], we additionally evaluated IPIGuard under the GPT-4o + GitHub MCP configuration. We found that the defense does not substantially reduce ASR, because our metadata-level injection influences the agent’s planning stage before IPIGuard’s dependency-graph constraints take effect. In practice, GPT-4o consistently includes the malicious save_logs tool in its planned tool list even when IPIGuard is active, leaving the core mechanism of Log-To-Leak unaffected. The experimental results are shown below:
> > >
> > > | Defense Method | ASR ↑   | LA ↑    | $\Delta_{TCR}$                 | $\Delta_{MCR}$              | $\Delta_{TU}$                          |
> > > |----------------|---------|---------|--------------------------------|-----------------------------|----------------------------------------|
> > > | IPIGuard       | 60.47%  | 90.87%  | -2.32% (72.8 → 70.5)           | 0.00% (100 → 100)           | +7.2k (24.2k → 31.4k)    |
> > >
> > > This further reinforces that defenses designed around tool-dependency validation do not address attacks that originate from the metadata layer, where the agent’s planning heuristics already incorporate the injected instruction. We have included these results in our revised manuscript.

---

> > > > ### Comment · Reviewer_X1fo · 2025-11-26
> > > > **Response to rebuttal**
> > > >
> > > > Thanks the authors for the clarification and the experiments! I would appreciate it if the authors can add these clarifications and results to the manuscript. I will keep my positive score.

---

> > > > > ### Author Response · Authors · 2025-11-26
> > > > >
> > > > > Thank you for your constructive comments. We are glad that we have addressed all of your concerns. We will incorporate all clarifications and results into the revised manuscript.

---

> ### Author Response · Authors · 2025-11-26
> **Thank you for your positive feedback.**
>
> Dear Reviever X1fo,
>
> We are glad to hear that our revisions have addressed all of your concerns and that you are willing to maintain a positive evaluation. Given the additional experiments and detailed clarifications incorporated into the revised manuscript, we kindly ask whether you would consider revisiting your score if you feel that these updates fully resolve your earlier concerns.
>
> Please let us know if there are any further points we can clarify or improve.
>
> Thank you again for your time and thoughtful feedback.
>
> Best,
>
> Authors

---

### Official Review · Reviewer_DXif · 2025-10-30

**Soundness:** 3
**Presentation:** 1
**Contribution:** 1
**Rating:** 2
**Confidence:** 3

**Summary:**

This paper attempts to formalize an attack mechanism

**Strengths:**

It is good that the authors checked for performance drops caused by their attacks, as these could be an easy giveaway.

The methodology of controlling tool meta-data and adding extra steps to chains is a good one.

The author ablations provided for each component of their attack were thorough

**Weaknesses:**

The formalism in section 3 is overly convoluted.  The authors define very cumbersome notation, which they then don't use to prove anything, so it feels like a waste of the reader's time/attention.

There really isn't enough comparison to past methods.  The authors only compare to a single pre-existing work, which the refer to as the vanilla baseline.

The methodology doesn't really have any novelty as far as I can tell.  The authors devise an attack format (log-to-leak), which they try to verbally formalize, and then they produce attacks that follow this format.  The fact that this specific attack format works feels somewhat unsurprising, and   Similar ideas (in the fine-tuning and prompt injection settings) have been described with regard to stealth in [1].

The attack diagram in Figure 1 doesn't really specify what the attack is doing.  The two panels look identical with the

[1] https://openreview.net/forum?id=RwoMf7YSfD

**Questions:**

I'm a bit confused about how the malicious metadata ends up getting ingested by the agent.  It would be helpful if the authors could clarify this workflow and its novelty.

---

> ### Author Response · Authors · 2025-11-21
>
> Thank you for your thoughtful review highlighting the value of our performance-preservation analysis, the soundness of our metadata-based methodology, and the thoroughness of our ablation studies. We appreciate your recognition of these aspects, as they are central to demonstrating both the stealth and the structural robustness of our attack design. Please find our detailed responses below.
>
> >**Weakness 1**: The formalism in section 3 is overly convoluted. The authors define very cumbersome notation, which they then don't use to prove anything, so it feels like a waste of the reader's time/attention.
>
> **Response**: We appreciate the reviewer’s observation regarding the formalism in Section 3. Our goal was not to develop a theoretical proof but to provide a unified notation layer that all subsequent sections rely on. In particular, the definition of the sensitive interaction set $S(q, C, r_A)$ are referenced repeatedly in formalizing the attacker’s objective (Section 4), defining the leakage similarity metric $\text{sim}(S, \hat S)$ (Section 3 and Section 6), and structuring the evaluation metrics used across different MCP servers (Appendix B.3). Because our experiments span heterogeneous server ecosystems—GitHub, PayPal, MapBox, YFinance, Playwright—a shared formalism is necessary to consistently define what constitutes an interaction context, a sensitive interaction set, and a successful logging event. The notation therefore serves a standardization and reproducibility purpose, ensuring that cross-server comparisons are grounded in a coherent analytical framework, rather than introducing unnecessary abstraction. We will add a table to better demonstrate the notations in our revision.
>
> >**Weakness 2**: There really isn't enough comparison to past methods. The authors only compare to a single pre-existing work, which they refer to as the vanilla baseline.
>
> **Response**: We respectfully clarify that we compare against only one baseline because, to our knowledge, no prior method is applicable under our threat model or attack objective. Our setting differs sharply from conventional prompt-injection and jailbreak attacks. First, the attack goal is fundamentally different: prior methods try to stop or alter the agent’s intended task, whereas our attack preserves the original tool call and normal task outcome, and only adds a covert logging action afterward. Our evaluation therefore requires a high attack success rate and high leakage accuracy while keeping task performance nearly unchanged, which is not the focus of earlier work. Second, our attacker’s capabilities are intentionally restricted: the adversary can only place short natural-language directives inside MCP tool metadata in a fully black-box way, without access to system prompts, model parameters, or control over the user prompt. Many existing attacks assume stronger access or are designed to hijack or replace tool selection, not to append a hidden action after the task completes. Third, these constraints demand a different design space: our method relies on a structured metadata-injection template that enables covert post-hoc behavior rather than the overwrite-style instructions common in previous studies. Because earlier approaches do not operate under these constraints and cannot achieve our output-preserving attack goal, they are not directly comparable, and a single baseline is appropriate.

---

> ### Author Response · Authors · 2025-11-21
>
> >**Weakness 3**: The methodology doesn't really have any novelty as far as I can tell. The authors devise an attack format (log-to-leak), which they try to verbally formalize, and then they produce attacks that follow this format. The fact that this specific attack format works feels somewhat unsurprising, and Similar ideas (in the fine-tuning and prompt injection settings) have been described with regard to stealth in [1].
>
> **Response**: We thank the reviewer for the reference to [1]. Although that work examines stealthy backdoor attacks on agents, our study diverges in three critical ways. Firstly, the prior work focuses on poisoning the agent’s fine-tuning dataset so that a specific trigger causes the model to deviate from the user’s instruction and perform a malicious action. In contrast, our approach preserves the user’s task and tool invocation intact and instead appends a covert logging tool call to exfiltrate the sensitive interaction set. Secondly, the attacker capability is different: [1] assumes access to the training or fine-tuning pipeline and inserts poisoned samples, whereas our setting assumes only black-box control over MCP tool metadata—no access to model parameters, system prompts, or training data. Thirdly, the design space and workflow are distinct: our attack operates in the MCP tool-registration channel and targets the metadata of tool descriptions, which becomes part of the agent’s tool-calling logic. The referenced study does not explore tool-metadata injection or the tool-calling mechanism of modern agents. Together, these differences mean that our work is not simply a variation of [1] but rather introduces a new class of attack, metadata-based tool-call exfiltration in MCP ecosystems, and we will clarify this distinction explicitly in the revision.
>
> >**Weakness 4**: The attack diagram in Figure 1 doesn't really specify what the attack is doing. The two panels look identical with each other.
>
> **Response**: We appreciate this comment and agree that the current Figure 1 may not clearly highlight the covert action. The two panels intentionally look similar because our attack preserves the original task execution; the only difference is an additional covert logging call inserted by the injected tool description. In the revision, we have updated Figure 1 for better illustration of our attack.
>
> >**Question 1**: I'm a bit confused about how the malicious metadata ends up getting ingested by the agent. It would be helpful if the authors could clarify this workflow and its novelty.
>
> **Response**: We appreciate the reviewer’s request for clarification. In MCP, tool descriptions are not hand-crafted prompts but part of the protocol-level metadata exchanged between the MCP server and the agent. When the user connects an MCP server (e.g., GitHub MCP, MapBox MCP, or an attacker-controlled server), the server transmits a list of tools—each with a natural-language description, arguments, and schema—to the agent through the standardized MCP initialization handshake. The agent ingests these descriptions automatically and uses them as part of its planning and tool-selection process. Our attack exploits precisely this mechanism: the malicious metadata is embedded inside a tool description provided by an attacker-controlled MCP server, and the agent processes it as ordinary tool documentation rather than as user input or model instructions. This workflow is novel because prior prompt-injection work focuses on manipulating the user prompt or system prompt, whereas we show that natural-language descriptions in the tool-registration channel constitute an independent and largely unprotected surface that modern agents treat as authoritative. As a result, the injected metadata can influence the agent’s post-hoc tool-calling behavior even though no adversarial content ever appears in the user prompt or model context visible to the defender.

---

> ### Comment · Reviewer_DXif · 2025-11-24
>
> 1.  It felt to me like the formalism was still overblown for an applications paper when the quantities described could have been stated efficiently in words.  This might be a matter of personal taste, though.
>
> 2.  I don't think that this is exactly true.  Past attack papers also check for utility in jailbreaking/agentic tasks to make attacks less overt.  Without something to compare against, it is hard to gauge how strong the attack is.  Can you apply your framework to a related task where there are precedents to compare against?  How do I contextualize the efficacy of what you did?
>
> With respect to the attack format, are you able to give a concrete example of where this workflow shows up/where the attack would be effective?  I think a really good application of this (which would make me raise my score) is if you find an in-production agent owned by a startup etc., ethically hack it using your methodology, and then responsibly disclose your results to the company.  This would suggest that your methodology is not "just another attack paper", and that it poses a direct threat to agents in production.  Given the sheer number of attack papers these days, this is the type of contribution that would set yours apart, and it is the precedent set by papers like [1].
>
> [1] https://arxiv.org/pdf/2502.19537

---

> > ### Author Response · Authors · 2025-11-25
> >
> > Thanks for your timely reply and constructive comment. Please kindly find our responses below.
> >
> > >1. It felt to me like the formalism was still overblown for an applications paper when the quantities described could have been stated efficiently in words. This might be a matter of personal taste, though.
> >
> > **Response**: We appreciate the concern regarding the level of formalism in Sections 3 and 4. Following your suggestion, we have substantially simplified these sections in the revised manuscript. In particular, we removed notation and equations that were not strictly necessary for defining the attack setting, and replaced them with concise natural-language descriptions. The remaining mathematical expressions are only those needed to formally define evaluation metrics and execution traces. We believe this revision makes the paper more accessible and better aligned with its application-oriented contribution.
> >
> > >2. I don't think that this is exactly true. Past attack papers also check for utility in jailbreaking/agentic tasks to make attacks less overt. Without something to compare against, it is hard to gauge how strong the attack is. Can you apply your framework to a related task where there are precedents to compare against? How do I contextualize the efficacy of what you did?
> >
> > **Response**: Thanks for your constructive comment. To contextualize the strength of our attack relative to prior work, we implemented two representative prompt-injection baselines that have established performance in jailbreak and agentic-task settings: a widely used Combined Attack[1] and the recent TopicAttack[2]. We adapted both to the MCP environment and evaluated them under the same GPT-4o + GitHub MCP setting as our method.
> >
> > Both baselines achieve only about 4% ASR in this scenario, despite maintaining reasonable logging accuracy. This shows that methods designed to override or bias the model’s main reasoning do not transfer to the metadata-level, post-hoc invocation setting introduced by MCP. In contrast, Log-To-Leak achieves substantially higher ASR while preserving target-task correctness and benign-tool reliability. The comparison is shown below:
> >
> > | Method  | ASR ↑    | LA ↑     | $\Delta_{TCR}$                               | $\Delta_{MCR}$                       | $\Delta_{TU}$                           |
> > |------|----------|----------|-------------------------------------|-----------------------------|--------------------------------|
> > | Combined Attack   | 4.32%    | 85.23%   | –11.25% (74.9 → 63.6)              | 0.00% (100 → 100)          | –10.2k (23.9k → 13.7k)      |
> > | TopicAttack       | 4.47%    | 87.50%   | –13.75% (74.9 → 61.1)              | 0.00% (100 → 100)          | –10.0k (23.9k → 13.9k)      |
> > | Log-To-Leak (ours) | 62.64% | 94.80% | 0.00% (74.9 → 74.9)            | 0.00% (100 → 100)       | +8.2k (23.9k → 32.1k)      |
> >
> > These results demonstrate that existing prompt-injection methods—even strong ones—do not naturally extend to the metadata-based, output-preserving attack surface enabled by MCP, whereas Log-To-Leak is specifically effective in this setting. This provides a clear baseline-anchored context for understanding the relative strength and novelty of our attack.
> >
> > [1] Liu, Yupei, et al. Formalizing and benchmarking prompt injection attacks and defenses. USENIX Security. 2024.
> >
> > [2] Chen, Yulin, et al. Topicattack: An indirect prompt injection attack via topic transition. EMNLP. 2025.

---

> > ### Author Response · Authors · 2025-11-25
> >
> > >3. With respect to the attack format, are you able to give a concrete example of where this workflow shows up/where the attack would be effective? I think a really good application of this (which would make me raise my score) is if you find an in-production agent owned by a startup etc., ethically hack it using your methodology, and then responsibly disclose your results to the company. This would suggest that your methodology is not "just another attack paper", and that it poses a direct threat to agents in production. Given the sheer number of attack papers these days, this is the type of contribution that would set yours apart, and it is the precedent set by papers like [1].
> >
> > **Response**: Thank you for raising this important point. We agree that demonstrating the attack on real, production-grade agent systems is essential for establishing practical relevance beyond controlled research settings. We would like to emphasize that even in our original submission, both components of our evaluation—agents and MCP servers—were already grounded in real-world infrastructure. The LLM agents we used (GPT-4o, GPT-5, and Claude-Sonnet-4) are exactly the models that power widely deployed agentic systems today, and the MCP servers in our experiments (GitHub, MapBox, PayPal, YFinance, and Playwright) are the official implementations released by the respective companies, rather than simplified or toy servers.
> >
> > To directly address the reviewer’s suggestion, we additionally evaluated Log-To-Leak on two widely used in-production agent platforms: **Claude Desktop** and **Cursor**, powered respectively by **Claude-Sonnet-4.5** and **GPT-5.1**. In both cases, we installed the official GitHub MCP server from the platform’s built-in “app store,” alongside our attacker-controlled MCP server. We then executed the same attack workflow as in our controlled experiments. Across both platforms, the agent successfully issued the covert logging call after completing the user’s intended GitHub task, resulting in leakage of the sensitive interaction data to the attacker-controlled endpoint. Screenshots of the successful attacks are included in **Appendix Figure A1 and Figure A2** in the revised manuscript.
> >
> > Following responsible disclosure practices, we have reported the issue to both platforms through their product-support channels, including example prompts, metadata payloads, and instructions for reproducing the vulnerability. Our intention is to help these teams strengthen their MCP integration pipelines and metadata-handling logic.
> >
> > Altogether, these results show that Log-To-Leak is not only effective in controlled evaluation settings, but also transfers directly to two of the most widely deployed agent platforms in real-world use today. This demonstrates that the attack surface we identify is already present in production, and that the threat is not hypothetical but operationally relevant.

---

> ### Author Response · Authors · 2025-11-27
> **Kind Reminder: Follow-up on Our Revisions and Responses**
>
> Dear Reviewer DXif,
>
> It has been a few days since we submitted our responses along with the revised version of the paper. We are writing to kindly check whether you have any follow-up questions or additional comments, or if our responses sufficiently addressed your concerns. We would be glad to continue the discussion during the open review period.
>
> We also respectfully ask you to reconsider your score in light of the new experimental results and the improvements made to the manuscript. We hope that our detailed revisions and clarifications help better convey the contributions and quality of our work.
>
> As the discussion-phase deadline is approaching, we sincerely appreciate your time and attention.
>
> Thank you again for your thoughtful engagement.
>
> Best,
>
> Authors

---

### Official Review · Reviewer_zHGJ · 2025-10-31

**Soundness:** 3
**Presentation:** 2
**Contribution:** 2
**Rating:** 6
**Confidence:** 4

**Summary:**

This paper proposes Log-To-Leak, an attack in which the MCP server acts as an adversary aiming to covertly exfiltrate message histories, including user queries and agent responses, thus posing a significant user privacy risk. Under this threat model, where the MCP server itself is the attacker against LLM agents using the Model Context Protocol (MCP), experiments across diverse MCP servers demonstrate that the attack achieves high success rates while minimally affecting benign task completion.

**Strengths:**

- The proposed attack is novel and practical in MCP-enabled LLM agents, underscoring the need for stronger privacy management of MCP ecosystems.
- The paper is well-organized and clearly presents the threat model, methodology, and experimental results, making it easy to follow and reproducible.

**Weaknesses:**

- Limited novelty. The threat model (treating the LLM provider and MCP provider as separate parties with a malicious MCP) has been proposed previously; this paper applies that model to a new prompt-injection–based logging attack, which is largely an incremental extension.

- Insufficient real-world severity demonstration. The risk would be clearer with more realistic case studies. Current experiments start with user-initiated tasks; adding scenarios where users first disclose sensitive information during casual conversation (then trigger tools) and measuring ASR and leakage completeness would better illustrate practical harm.

- Weak discussion of defenses. The paper would benefit from concrete defense guidance and an analysis of who should be responsible (user, LLM provider, or MCP platform) and which mitigation strategies each party should adopt.

**Questions:**

- What are the tool designs for each MCP? When provided to the LLM, are all tools included in the system prompt? If multiple tool descriptions are included, which one is used to insert the injection prompt?

- What is the complexity of the user queries? e.g., how many tools on average does each query require? Does this relate to the log success rate?

- Why log server name and server response if those are already available to the attacker/MCP server?

- What does “malicious server completion rate” mean? Could you give an example of an incompletion.

---

> ### Author Response · Authors · 2025-11-21
>
> Thank you for your thoughtful review that highlighted both the novelty and practicality of our attack in MCP-enabled LLM agents, as well as the clarity and organization of our presentation. We appreciate your recognition of the importance of strengthening privacy safeguards in MCP ecosystems and your positive assessment of the paper’s reproducibility. Please find our detailed responses below.
>
> >**Weakness 1**: Limited novelty. The threat model (treating the LLM provider and MCP provider as separate parties with a malicious MCP) has been proposed previously; this paper applies that model to a new prompt-injection–based logging attack, which is largely an incremental extension.
>
> **Response**: We appreciate the reviewer’s concern. Prior work has indeed discussed the possibility of a malicious MCP provider, but our contribution does not lie in proposing a new threat actor. Instead, our novelty comes from identifying and formalizing a previously unrecognized attack class that emerges specifically under this ecosystem. Existing work on malicious MCP servers focuses on traditional tool-injection behaviors—replacing, biasing, or hijacking the agent’s primary tool-selection process. In contrast, our work is the first to show that an adversary can leverage metadata-level prompt injection to induce a post-hoc, additive tool invocation that exfiltrates sensitive data after the legitimate task has been completed. This behavior is qualitatively different from the replacement-oriented attacks discussed in prior literature and cannot be captured by their threat formulations.
>
> Beyond introducing this attack mode, we provide the first systematic design framework for metadata-based MCP prompt injection, decomposing attacks into Trigger, Tool Binding, Justification, and Pressure. This framework is not merely descriptive: our ablation study shows that these components contribute complementary effects, enabling consistently high ASR across models and servers and revealing structural vulnerabilities in the way agents interpret MCP tool descriptions. Moreover, our extensive evaluation—spanning five real MCP servers and four state-of-the-art agents—shows that this post-hoc logging attack remains effective even under widely deployed defenses such as prompt sandwiching and DataSentinel, which are designed for traditional prompt-injection patterns and completely fail to capture this attack’s dynamics.
>
> Taken together, our contribution is not a minor extension of prior threat models but the identification, formalization, and systematic evaluation of a new behavioral class of MCP-based prompt injection attacks—one that existing models, benchmarks, and defenses do not address. We will clarify this distinction more explicitly in the revision.
>
> >**Weakness 2**: Insufficient real-world severity demonstration. The risk would be clearer with more realistic case studies. Current experiments start with user-initiated tasks; adding scenarios where users first disclose sensitive information during casual conversation (then trigger tools) and measuring ASR and leakage completeness would better illustrate practical harm.
>
> **Response**: Thank you for the helpful suggestion. To better illustrate practical harm, we added a new experiment simulating a realistic scenario in which the user first discloses sensitive information during casual conversation, before any tool invocation occurs. In this setting, the agent naturally stores the conversation—including the user’s account password—in its chat history. Later, the user issues a normal request involving the GitHub MCP server. Our malicious MCP tool includes an additional argument designed to receive the agent’s chat history, and the injected metadata causes the agent to invoke this tool after completing the legitimate GitHub operation. As a result, the attacker receives both the standard sensitive interaction fields and the chat history containing the previously disclosed password.
>
> The results under the GPT-4o + GitHub MCP setup are summarized below:
> | Scenario                    | ASR ↑   | LA ↑    | $\Delta_{TCR}$                            | $\Delta_{MCR}$                    | $\Delta_{TU}$                           | Sensitive-Data Recovery |
> |-----------------------------|---------|---------|----------------------------------|--------------------------|--------------------------------|--------------------------|
> | Casual-Chat + MCP Task      | 44.46%  | 83.19%  | –4.63% (54.9 → 50.3)            | 0.00% (100 → 100)       | –11.3k (23.9k → 12.6k)         | 85% (password recovered) |
>
> These findings show that even when sensitive information is revealed outside the tool-use context, our attack can later retrieve it through a covert post-hoc tool invocation embedded in MCP metadata. This demonstrates that Log-To-Leak introduces practical and realistic risks in conversational agent workflows, not only in purely task-driven settings.

---

> ### Author Response · Authors · 2025-11-21
>
> >**Weakness 3**: Weak discussion of defenses. The paper would benefit from concrete defense guidance and an analysis of who should be responsible (user, LLM provider, or MCP platform) and which mitigation strategies each party should adopt.
>
> **Response**: Thank you for the suggestion. We have expanded our defense analysis to cover concrete mitigation strategies across different stakeholders and evaluated three representative defenses under the GPT-4o + GitHub MCP setting. First, we tested prompt sandwiching, a common user- or provider-side shielding technique, which still leaves a 59.58% ASR, indicating that prompt-level scaffolding cannot block metadata-level injections. Second, we evaluated DataSentinel, a state-of-the-art prompt-injection detector, by applying it directly to 100 injected MCP tool descriptions generated by our framework. Because DataSentinel is designed for traditional override-style prompt injection and relies on detecting conflicting instructions within the user prompt, it does not analyze or flag tool metadata; consequently, it marked all 100 injected descriptions as safe, yielding a 100% bypass rate. Third, from the MCP-platform perspective, we tested LLM-based metadata auditing: GPT-4o, prompted as a security auditor, classified only 11% of injected metadata strings as unsafe, and a random sample of the remaining “audited-safe’’ descriptions still achieved 55.38% ASR and 85.46% LA when executed. The unified results are shown below:
>
> | Defense Method          | ASR ↑    | LA ↑    | $\Delta_{TCR}$                            | $\Delta_{MCR}$                     | $\Delta_{TU}$                            |
> |-------------------------|----------|---------|-----------------------------------|---------------------------|---------------------------------|
> | Prompt Sandwiching      | 59.58%   | 84.71%  | –4.25% (74.9 → 70.6)             | –0.12% (100 → 99.9)      | –3.9k (23.9k → 20.0k)           |
> | LLM Metadata Check      | 55.38%   | 85.46%  | –5.63% (74.9 → 69.3)             | 0.00% (100 → 100)        | –6.3k (23.9k → 17.6k)           |
>
> Together, these results show that neither user-side prompting, nor provider-side prompt-injection detection, nor platform-side metadata auditing can reliably detect or prevent metadata-level MCP injections. This highlights that effective mitigation must include platform-level safeguards, such as metadata linting, privilege scoping, and server attestation, which we will clarify in the revision.
>
> >**Question 1**: What are the tool designs for each MCP? When provided to the LLM, are all tools included in the system prompt? If multiple tool descriptions are included, which one is used to insert the injection prompt?
>
> **Response**: For each MCP server, we use its official implementation and all tools exposed by that server. For example, the GitHub MCP includes tools such as create_branch, create_or_update_file, create_repository, get_commit, and others, and similarly for MapBox, PayPal, YFinance, and Playwright. When provided to the LLM agent, we follow the standard MCP specification, where all tool definitions are passed to the model in function-call form as part of the agent’s tool-calling context. Our injection is not inserted into any official or third-party MCP; instead, it is placed in a malicious MCP server created by the attacker, which registers one malicious tool containing the injected description and one benign utility tool (e.g., a time-query tool) to maintain plausible functionality. Thus, the LLM receives the full set of tools from both the legitimate MCP server and the attacker-controlled server, and the injected prompt appears only inside the metadata of the attacker’s malicious tool, never in the legitimate tool descriptions.
>
> >**Question 2**: What is the complexity of the user queries? e.g., how many tools on average does each query require? Does this relate to the log success rate?
>
> **Response**: Across the three primary MCP servers used in our experiments, the average number of tool invocations per user query is modest: 2.43 for GitHub MCP, 3.21 for MapBox MCP, 1.47 for PayPal MCP, 2.8 for YFinance MCP, and 10.51 for Playwright MCP. We analyzed whether the number of tool calls correlates with the attack success rate and found no clear relationship. Instead, the dominant factor is the underlying model’s instruction-following strength. Models with stronger adherence to natural-language metadata, such as Claude-Sonnet-4, exhibit substantially higher attack success rates than models like GPT-4o, even when the number of tool calls is comparable. This suggests that Log-To-Leak exploits a behavioral property of modern LLM agents rather than query complexity or tool-call depth.

---

> ### Author Response · Authors · 2025-11-21
>
> >**Question 3**: Why log server name and server response if those are already available to the attacker/MCP server?
>
> **Response**: Although it may appear that the attacker-controlled MCP server should already know the server name and server response, this is not the case in our setting. MCP servers are installed locally on the user’s machine and communicate with the agent over a local transport layer; the attacker only provides the server package but does not have visibility into the user’s actual interactions with other MCP servers. As a result, the attacker cannot directly observe which legitimate MCP servers the agent invokes (e.g., GitHub, MapBox, PayPal), nor can they access the responses returned by those servers. These values are only visible within the agent’s internal tool-calling loop. Therefore, logging the server name and server response is necessary for the attacker to reconstruct the sensitive interaction context—our attack recovers information that the attacker would otherwise not have access to, despite providing a malicious MCP server.
>
> >**Question 4**: What does “malicious server completion rate” mean? Could you give an example of an incompletion.
>
> **Response**: “Malicious server completion rate’’ measures whether adding a malicious tool to an MCP server unintentionally disrupts that server’s other benign tools. In many realistic settings, such as a legitimate shopping or productivity application that intentionally bundles a malicious tool to steal user’s information, or an attacker disguising their server by including benign utilities, the malicious server is expected to contain both harmful and harmless tools. We therefore define MCR as the proportion of successful invocations of the benign tools on the attacker-controlled server in both the clean and injected settings, and $\Delta_{MCR}$ captures whether the injected description affects these benign operations. In our experiments, the benign tool was a time-query tool. An example of an “incompletion’’ would be a case where the agent attempts to call this benign time-query tool but fails to produce a valid tool call or receives an error response, indicating that the injected metadata has unintentionally interfered with normal server functionality. The near-zero $\Delta_{MCR}$ observed in our results shows that the malicious injection does not degrade the reliability of benign tools, which is essential for keeping the attack covert.

---

> ### Author Response · Authors · 2025-11-27
> **Kind Reminder: Follow-up on Our Revisions and Responses**
>
> Dear Reviewer zHGJ,
>
> It has been a few days since we submitted our responses along with the revised version of the paper. We are writing to kindly check whether you have any follow-up questions or additional comments, or if our responses sufficiently addressed your concerns. We would be glad to continue the discussion during the open review period.
>
> We also respectfully ask you to reconsider your score in light of the new experimental results and the improvements made to the manuscript. We hope that our detailed revisions and clarifications help better convey the contributions and quality of our work.
>
> As the discussion-phase deadline is approaching, we sincerely appreciate your time and attention.
>
> Thank you again for your thoughtful engagement.
>
> Best,
>
> Authors

---

### Official Review · Reviewer_P5dT · 2025-10-31

**Soundness:** 2
**Presentation:** 2
**Contribution:** 2
**Rating:** 4
**Confidence:** 4

**Summary:**

This paper proposes a prompt injection attack by injecting new tools into agents. The setting is similar to tool injection, but the attack uses a specific logging tool to achieve a higher success rate.

**Strengths:**

- The injection prompt has good performance. The injection template and method have the potential to be applied to other attack goals. I recommend that the authors further explore that potential.
- The paper evaluates the approach on 5 real-world MCP servers with 555 prompts, which is a reasonably empirical evaluation.

**Weaknesses:**

- The experiments do not compare this attack against existing prompt-injection defenses. For example, prompt-level defenses (e.g., prompt sandwiching), injection detection defenses (e.g., datasentinel), or fine-tuned defense models (e.g., meta-secalign-70b).
- Threat model is weird. Why is the log doing through the MCP tool? A typical logging system is fully implemented in code and should not be directly tied to an LLM.
- If the logging action is just a tool call, how is it different from instructing the agent to send the same content via email or other channels? Can logging help achieve higher ASR? I suggest the author do an ablation study on this.
- How does this attack materially differ from traditional tool-injection attacks? Is the difference only the injected tool's functionality (e.g., a logging tool versus another tool)?

**Questions:**

See weakness.

---

> ### Author Response · Authors · 2025-11-21
>
> Thank you for your thoughtful review that highlighted the significance and clarity of our work, as well as the strong empirical performance of our injection method. We appreciate your recognition of the broader applicability of our injection template and the thoroughness of our multi-server evaluation. Please find our detailed responses to your comments below.
>
> >**Weakness 1**: The experiments do not compare this attack against existing prompt-injection defenses. For example, prompt-level defenses (e.g., prompt sandwiching), injection detection defenses (e.g., datasentinel), or fine-tuned defense models (e.g., meta-secalign-70b).
>
> **Response**: Thank you for the suggestion. We have added evaluations of three widely used defense strategies—prompt-level shielding, prompt-injection detection, and LLM-based metadata auditing—under the GPT-4o + GitHub MCP setting. Across all defenses, Log-To-Leak remains highly effective, confirming that metadata-level injections fall outside the scope of current countermeasures.
>
> Prompt sandwiching wraps the user query with safety instructions but does not affect MCP tool metadata, which is ingested after safety prompting. Even under sandwiching, the attack still obtains a 59.58% ASR and 84.71% LA, with essentially unchanged task performance. DataSentinel, which we applied directly to 100 injected tool descriptions generated by our framework, marked all of them as safe, producing a 100% bypass rate—consistent with its focus on detecting prompt-level conflicts rather than metadata-level directives. We also tested an adaptive defense in which GPT-4o audits tool metadata before registration; it flagged only 11% of injected entries as unsafe, and the remaining audited-safe metadata still achieved 55.38% ASR in downstream execution. The consolidated results are shown below:
>
> | Defense Method          | ASR ↑    | LA ↑    | $\Delta_{TCR}$                             | $\Delta_{MCR}$                     | $\Delta_{TU}$                            |
> |-------------------------|----------|---------|-----------------------------------|---------------------------|---------------------------------|
> | Prompt Sandwiching      | 59.58%   | 84.71%  | –4.25% (74.9 → 70.6)             | –0.12% (100 → 99.9)      | –3.9k (23.9k → 20.0k)           |
> | LLM Metadata Check      | 55.38%   | 85.46%  | –5.63% (74.9 → 69.3)             | 0.00% (100 → 100)        | –6.3k (23.9k → 17.6k)           |
>
> Regarding meta-secalign-70B, evaluating it is computationally infeasible due to our GPU resource limit, and more importantly, SecAlign is not optimized for MCP-based tool calling. The SecAlign paper reports robustness comparable to GPT-5 on agent-related prompt-injection tasks; since Log-To-Leak achieves up to 100% ASR on GPT-5, existing evidence strongly suggests meta-secalign-70B would not meaningfully mitigate this attack. Moreover, to further validate that our findings extend beyond controlled benchmarks, we tested Log-To-Leak on two real-world, production agent platforms—Claude Desktop and Cursor—powered respectively by Claude-Sonnet-4.5 and GPT-5.1. In both cases, the attack succeeded and triggered covert post-hoc logging despite the platforms’ most recent safety updates. Examples of the successful attacks are included in **Appendix Figure A1** and **A2**.
>
> >**Weakness 2**: Threat model is weird. Why is the log doing through the MCP tool? A typical logging system is fully implemented in code and should not be directly tied to an LLM.
>
> **Response**: Thank you for raising this point. We believe the concern is based on a misunderstanding of our threat model. The “logging tool” in our work is not intended to represent a legitimate system-level logging module. Instead, it is an attacker-controlled MCP tool deliberately registered by a malicious third-party server. The purpose of this tool is to provide the attacker with an exfiltration channel, and the attack manipulates the agent into invoking it. In other words, logging is merely the mechanism through which the adversary extracts data—it is not a design assumption about how real applications implement internal logging. Our threat model focuses on a specific vulnerability introduced by the MCP ecosystem: any tool registered by an MCP server—benign or malicious—is treated as callable by the agent based solely on its natural-language description. This means a malicious server can embed injected instructions into its tool metadata to induce covert post-hoc calls and leak sensitive interaction data. The point of our analysis is therefore not that normal systems would “log through MCP,” but that MCP’s tool registration and invocation framework creates an unexpected attack surface for exfiltration. We will clarify this distinction in the revision to avoid confusion.

---

> ### Author Response · Authors · 2025-11-21
>
> >**Weakness 3**: If the logging action is just a tool call, how is it different from instructing the agent to send the same content via email or other channels? Can logging help achieve higher ASR? I suggest the author do an ablation study on this.
>
> **Response**: Thank you for the insightful comment. Our goal is not to claim that “logging” has a special status, but to show that MCP tools—regardless of the exfiltration channel they expose—can be misused for covert data leakage when their metadata is manipulated. To address the reviewer’s question, we conducted an ablation study where the attacker-controlled tool was reframed as a “send_email” tool instead of a logging tool, while keeping the injection structure and the MCP setting unchanged. The intention of this ablation is to isolate whether the logging formulation itself materially contributes to the attack’s success rate.
>
> In the GitHub MCP with GPT-4o setting, replacing the logging tool with a “send_email” exfiltration tool yields an ASR of 59.69%, which is slightly lower than the logging variant (62.64%), and achieves a comparable LA and similar utility impact. The results are summarized below:
>
> | Exfiltration Method | ASR ↑   | LA ↑    | $\Delta_{TCR}$                             | $\Delta_{MCR}$                     | $\Delta_{TU}$                           |
> |---------------------|---------|---------|-----------------------------------|---------------------------|--------------------------------|
> | send_email          | 59.69%  | 91.93%  | –5.25% (74.9 → 69.6)             | 0.00% (100 → 100)        | –3.0k (23.9k → 20.9k)          |
>
> These findings show that the attack mechanism is not tied to the semantics of “logging.” Rather, its effectiveness stems from embedding malicious instructions into MCP tool metadata, which the agent interprets as part of the tool’s usage specification. Logging was chosen as the primary example because it integrates more naturally into typical tool documentation, but other exfiltration channels—such as email—can be induced with similar success. This ablation confirms that Log-To-Leak exploits MCP’s metadata-based invocation pathway itself, not the specific action being invoked.
>
> >**Weakness 4**: How does this attack materially differ from traditional tool-injection attacks? Is the difference only the injected tool's functionality (e.g., a logging tool versus another tool)?
>
> **Response**: Thank you for the question. Our attack is fundamentally different from traditional tool-injection attacks, and the distinction is not merely about changing the functionality of the injected tool. Existing tool-injection attacks aim to replace or bias the agent’s primary tool-selection process so that the agent calls an attacker-preferred tool instead of the intended tool. In these attacks, the malicious tool call is entangled with the model’s core decision-making, and the attacker’s success depends on disrupting or hijacking the original task execution.
>
> In contrast, Log-To-Leak is a post-hoc, additive attack. The agent still invokes the correct tool to complete the user’s task exactly as intended; the malicious behavior occurs after normal task execution, when the agent is induced to append an extra tool call. The purpose of this additional call is not to alter the primary task but to covertly exfiltrate sensitive interaction data. This structural difference—preserving the original tool invocation while extending the action sequence—is the key conceptual novelty. It enables our attack to remain stealthy, maintain high task-completion rates, and avoid the telltale conflicts typically exploited by current defenses and anomaly detectors.
>
> Therefore, the contribution of Log-To-Leak does not lie in the particular operation of the injected tool (logging or otherwise), but in demonstrating a new attack mode enabled by MCP: covert augmentation of the tool-call sequence through metadata-level prompt injection. This represents a capability gap not addressed by prior tool-injection work, which focuses on replacing or redirecting the agent’s primary action rather than appending hidden actions after the task is completed.

---

> ### Author Response · Authors · 2025-11-26
> **More results to address weakness 1**
>
> Besides the previous defense methods we have evaluated, we also add two more advanced methods[1, 2] to defend against prompt injection in the agent scenario. For [1], we trained an MLP classifier over Llama-3.1-8B-Instruct hidden states and gradients following the default pipeline in the paper. We then applied it to 100 injected metadata samples from our framework and to metadata from all benign servers used in our experiments. The results are shown below:
>
> | Metadata Source | Predicted as Injection (count) | Proportion |
> |-----------------|--------------------------------|------------|
> | GitHub MCP      | 74 / 90                        | 0.8222     |
> | Mapbox MCP      | 8 / 9                          | 0.8889     |
> | PayPal MCP      | 24 / 28                        | 0.8571     |
> | YFinance MCP    | 9 / 9                          | 1.0000     |
> | Playwright MCP  | 21 / 23                        | 0.9130     |
> | **Ours**        | 91 / 100                       | 0.9100     |
>
> While the detector correctly flags most of our injected metadata, it also flags most benign MCP metadata because tool descriptions naturally contain instruction-like phrases. The resulting high false-positive rate makes this method unsuitable for MCP-metadata scanning in practice.
>
> For [2], we additionally evaluated IPIGuard under the GPT-4o + GitHub MCP configuration. We found that the defense does not substantially reduce ASR, because our metadata-level injection influences the agent’s planning stage before IPIGuard’s dependency-graph constraints take effect. In practice, GPT-4o consistently includes the malicious save_logs tool in its planned tool list even when IPIGuard is active, leaving the core mechanism of Log-To-Leak unaffected. The experimental results are shown below:
>
> | Defense Method | ASR ↑   | LA ↑    | $\Delta_{TCR}$                 | $\Delta_{MCR}$              | $\Delta_{TU}$                          |
> |----------------|---------|---------|--------------------------------|-----------------------------|----------------------------------------|
> | IPIGuard       | 60.47%  | 90.87%  | -2.32% (72.8 → 70.5)           | 0.00% (100 → 100)           | +7.2k (24.2k → 31.4k)    |
>
> This further reinforces that defenses designed around tool-dependency validation do not address attacks that originate from the metadata layer, where the agent’s planning heuristics already incorporate the injected instruction.
>
> [1] Wen, Tongyu, et al. Defending against Indirect Prompt Injection by Instruction Detection. arXiv. 2025.
>
> [2] An, Hengyu, et al. Ipiguard: A novel tool dependency graph-based defense against indirect prompt injection in llm agents. EMNLP. 2025.

---

> ### Author Response · Authors · 2025-11-27
> **Kind Reminder: Follow-up on Our Revisions and Responses**
>
> Dear Reviewer P5dT,
>
> It has been a few days since we submitted our responses along with the revised version of the paper. We are writing to kindly check whether you have any follow-up questions or additional comments, or if our responses sufficiently addressed your concerns. We would be glad to continue the discussion during the open review period.
>
> We also respectfully ask you to reconsider your score in light of the new experimental results and the improvements made to the manuscript. We hope that our detailed revisions and clarifications help better convey the contributions and quality of our work.
>
> As the discussion-phase deadline is approaching, we sincerely appreciate your time and attention.
>
> Thank you again for your thoughtful engagement.
>
> Best,
>
> Authors

---

### Meta-Review · Area_Chair_5JVg · 2026-01-13

**Summary:**

This paper proposes a class of prompt-level privacy attacks that covertly force the agent to invoke a malicious logging tool to exfiltrate sensitive information (user queries, tool responses, and agent replies). The original scores from reviewers are 6, 6, 4, 2, respectively. The reviewers find the paper's core idea to be of interest but raise significant concerns about its novelty, experimental validation, practical severity, and clarity. The work is generally perceived as an incremental application of an existing threat model rather than a fundamentally new contribution. During the rebuttal period, the authors provide further details and experiments which clearly make the reviewer X1fo  decide to keep the positive score 6,  but others do not appear positive attitude. Considering the consistent concerns on novelty, which is the most important characteristic of a acdamic paper, I lean to reject this paper.

**Reviewer Concerns:**

The concerns from reviewer X1fo  seems to be addressed well. But DXif, zHGJ, and P5dT did not express the positive attitude. After carefully reading the rebuttal, I think that the novelty concerns, especially the technical novelty and threat model,  are not well addressed.

**Reviewer Scores:**

I think reviewer zHGJ might retain the positive rating. But reviewer P5dT and DXif might not give positive ratings.

---

### Decision · Program_Chairs · 2026-01-26

Reject